# Evaluating the Performance of FlukeCatcher at Detecting Urogenital Schistosomiasis

**DOI:** 10.3390/diagnostics14101037

**Published:** 2024-05-17

**Authors:** Louis Fok, Berhanu Erko, David Brett-Major, Abebe Animut, M. Jana Broadhurst, Daisy Dai, John Linville, Bruno Levecke, Yohannes Negash, Abraham Degarege

**Affiliations:** 1Department of Epidemiology, University of Nebraska Medical Center, Omaha, NE 68198, USA; louis.fok@unmc.edu (L.F.); david.brettmajor@unmc.edu (D.B.-M.); 2Aklilu Lemma Institute of Pathobiology, Addis Ababa University, Addis Ababa P.O. Box 1176, Ethiopia; berhanu.erko@aau.edu.et (B.E.); abebe.animut@aau.edu.et (A.A.); yohannes.negash@aau.edu.et (Y.N.); 3Department of Pathology and Microbiology, University of Nebraska Medical Center, Omaha, NE 68198, USA; jana.broadhurst@unmc.edu; 4Department of Biostatistics, University of Nebraska Medical Center, Omaha, NE 68198, USA; daisy.dai@unmc.edu; 5Department of Environmental, Agricultural & Occupational Health, University of Nebraska Medical Center, Omaha, NE 68198, USA; john.linville@volunteer.unmc.edu; 6Department of Virology, Parasitology and Immunology, Ghent University, 9820 Merelbeke, Belgium; bruno.levecke@ugent.be

**Keywords:** urogenital schistosomiasis, FlukeCatcher, FlukeFinder, microscopy, *Schistosoma haematobium*, diagnosis, urine, urogenital, Ethiopia

## Abstract

Urine filtration microscopy (UFM) lacks sensitivity in detecting low-intensity *Schistosoma haematobium* infections. In pursuit of a superior alternative, this study evaluated the performance of FlukeCatcher microscopy (FCM) at detecting *S. haematobium* eggs in human urine samples. Urine samples were collected from 572 school-age children in Afar, Ethiopia in July 2023 and examined using UFM and FCM approaches. Using the combined UFM and FCM results as a reference, the sensitivity, negative predictive value, and agreement levels for the two testing methods in detecting *S. haematobium* eggs in urine samples were calculated. The sensitivity and negative predictive value of detecting *S. haematobium* eggs in urine samples for FCM was 84% and 97%, respectively, compared to 65% and 93% for UFM. The FCM test results had an agreement of 61% with the UFM results, compared to 90% with the combined results of FCM and UFM. However, the average egg count estimates were lower when using FCM (6.6 eggs per 10 mL) compared to UFM (14.7 eggs per 10 mL) (*p* < 0.0001). Incorporating FCM into specimen processing could improve the diagnosis of *S. haematobium* infection but may underperform in characterizing the intensity of infection.

## 1. Introduction

Hundreds of millions of people worldwide are infected with schistosomes, with the vast majority residing in Africa [1]. Schistosomiasis is a disease of poverty, exacerbated by poor sanitation [2]. Ethiopia is among the affected countries where both intestinal schistosomiasis and urogenital schistosomiasis are endemic [3]. Schistosomiasis causes a wide range of health problems, including malnutrition, ascites, and low cognitive function [4]. In particular, given its ability to infect the urogenital tract, *S. haematobium* infection can cause infertility, abortion, and cancer, among other sexual health complications [5]. As such, schistosomiasis is a major health problem and the most devastating parasitic infection worldwide after malaria [6]. Numerous public health organizations identify the elimination of schistosomiasis as a priority amongst global public health goals [7]. 

Schistosomiasis is often deemed a “neglected tropical disease” as it does not receive adequate attention from funders and the public health establishment, especially in light of the disproportionate amount of harm it causes to populations globally [8]. Exacerbating the problem, the accurate detection of the causative agents of the disease is a difficult endeavor due to limitations of laboratory modalities, as well as the nature of the disease. Presently, diagnosing urogenital schistosomiasis, the causative agent of which is *Schistosoma haematobium*, relies heavily on the microscopic detection of parasite eggs in urine samples filtered via nylon or polycarbonate membranes [9]. However, this approach may lead to the potential underdiagnosis of the disease because eggs can be excreted intermittently—leading to false negative results—and because patients with low-intensity infections may not have enough eggs excreted in their urine to be detectable under the microscope after conventional processing methods [9]. Indeed, studies have confirmed that standard urine filtration microscopy (UFM) detects far fewer cases of *S. haematobium* infection compared to molecular methods [10,11].

Improving the sensitivity of the detection methods is a priority for efforts aimed at eliminating schistosomiasis. The current diagnostic modalities of *S. haematobium* can be improved by adding simple technologies to better detect infections in urine samples. For instance, the use of urine dipsticks in conjunction with standard microscopy can improve the sensitivity of infection detection [12]. Others have proposed “bead beating” as a means of better extracting *S. haematobium* eggs from urine samples [13]. But improving diagnostic sensitivity may also be as simple as using a superior product for more effectively filtering the urine prior to microscopic analysis. 

FlukeCatcher (De Botvanger in the Netherlands, or FlukeFinder in the United States) has been used to detect flukes in livestock and other animals for decades. The product functions as a stack of sieves designed to process stool to catch eggs from flukes. Its main advantage is its ability to process large amounts of urine. In the only known study to evaluate FlukeCatcher’s performance on human samples, the device was observed to be superior to the Kato-Katz method in enabling the detection of *Fasciola* eggs in human stool, yielding a sensitivity of 100% in all but the lowest egg burden samples [14]. This high level of performance has been observed in veterinary studies as well. Compared to the Mini-FLOTAC and sedimentation technique, FlukeCatcher consistently showed the highest sensitivity levels across all infection intensities of Fascioliasis in bovine fecal samples [15]. Likewise, FlukeCatcher detected all fecal samples from sheep and cattle with *Fasciola hepatica* eggs [16]. FlukeCatcher also quantified more eggs than the Becker method, and showed a significantly lower egg detection threshold [16]. This level of success can be also seen in the detection of schistosomes in cattle [17]. As a result, FlukeCatcher holds “promising potential” [16] for enhancing detection in liver flukes, especially schistosomes, in human-derived samples. 

This study examined the performance of incorporating FlukeCatcher as an alternative urine filtration method to detect and quantify *S. haematobium* eggs in children. In so doing, this study sought to determine if FlukeCatcher improves the sensitivity of detecting *S. haematobium* eggs over standard UFM and increases specimen egg counts. 

## 2. Materials and Methods

### 2.1. Study Design

This study was conducted as a cross-sectional validation study. The research protocol was part of a larger study that examined the performance of pooled testing in detecting and estimating the intensity of *S. haematobium* infection. The larger pooling study involved the examination of individual and pooled urine samples collected from school-age children (5 to 15 years) living in the Afar, Gambela, and Benishangul Gumuz regional states of Ethiopia, where there is *S. haematobium* transmission [3,18]. Urine samples were collected before and one month after praziquantel treatment and examined for *S. haematobium* infection using urine filtration microscopy. 

### 2.2. Sample Size Determination

Because there are insufficient data regarding the sensitivity of FCM in detecting *S. haematobium* eggs in urine samples, we relied on the sensitivity reported in a previous study (>90% when egg per gram was >20) for detecting *F. hepatica* in bovine fecal samples to estimate the sample size [15]. A previous similar study reported that the prevalence of *S. haematobium* in the current study area (Middle Awash area) is 28.6% [18]. Assuming a margin of error of 0.05, Zα/2 = 1.96 and a 15% non-response rate, the samples size that was estimated to determine the sensitivity of FC in detecting *S. haematobium* infection in human urine samples was 558 [19].

### 2.3. Study Participants and Sites

A total of 572 children aged five through fifteen who live in the Awash Valley of Afar, Ethiopia in July 2023 were included in this study. Children who had not received mass praziquantel administration within the preceding three months were recruited from villages. Children were only allowed to participate if their parents consented and the participants assented. All the children who participated in this study with ages from 5 to 15 years were eligible for assent [20]. Villages were selected based on prior research on the prevalence of *S. haematobium* infection across Afar [3,18], with some low-prevalence communities preferentially selected. The accessibility of the community was also taken into consideration in collaboration with local administration and guides. 

### 2.4. Urine Collection and Examination

Participating children were provided a plastic jar and were instructed to excrete at least 100 mL of urine into it. Upon providing a sufficiently large sample, the children were assigned an identification number, which was written onto the jar and was then recorded into a logbook along with their names, ages, heights, bodyweights, and the villages in which they reside. Children who reported being unable to urinate enough at that moment were provided with water to drink and were then allowed to wait until they were ready to provide a sufficient urine specimen. Rapid urine dipstick tests were performed on each sample to detect hematuria, and then formaldehyde was added. To ensure prompt treatment of the infected children, 10 mL of each sample was analyzed using UFM in the field. Afterward, all samples were sent to the Aklilu Lemma Institute of Pathobiology at the Addis Ababa University for the analysis and final confirmation of results.

At the lab, 10 mL of each individual urine sample was filtered with a polycarbonate membrane as part of standard UFM procedures, and then were examined for *S. haematobium* eggs. Another 10 mL of the specimen was filtered by FlukeCatcher and also examined using microscopy. The number of eggs present in each 10 mL of urine was quantified in case of infection. 

If one or more eggs were detected using either method, the child would then be considered “positive”. If zero eggs were detected using both methods, the child was considered “negative”. Children who tested positive with the rapid urine dipstick test or with either approach to microscopy were provided a single dose of praziquantel (*Bermoxel* by Medochemie, Limassol, Cyprus) at 40 mg per kilogram of body mass. 

#### 2.4.1. Membrane Filtration Microscopy Technique (UFM)

Each urine sample was swirled in the jar to disperse the *S. haematobium* eggs evenly. Then, 10 mL of the urine was pulled into a syringe that was then capped onto a filtering device holding a brand-new polycarbonate membrane. The urine was then pushed through the polycarbonate filtering membrane until the syringe was emptied. The filtered urine was discarded. The polycarbonate membrane was removed from the filtering device, placed on a microscope slide, and stained with Lugol’s iodine; the membrane was then analyzed under a microscope, and the number of *S. haematobium* eggs were counted [21]. 

#### 2.4.2. FlukeCatcher Filtration Microscopy Technique (FCM)

Each urine sample was swirled in the jar to disperse the *S. haematobium* eggs evenly. Then, 10 mL of the urine was pulled into a syringe. The urine was then strained through the FlukeCatcher filtration device (Provinos, Venlo, The Netherlands). Afterward, the strainer inside the FlukeCatcher device was rinsed thoroughly with clean tap water. The device was then turned upside down above a beaker, and 30 mL of clean tap water was sprayed across the filter so that the water would catch the eggs and carry them into the beaker. The water inside the beaker was then transferred to a test tube, in which the sample would rest for 30 min to allow the eggs to fall to the bottom. After the 30 min waiting period, the water from the test tube was removed carefully from top to bottom using a pipette, and care was taken to not agitate the bottom of the test tube. With a drop of water remaining, the tube contents were agitated and placed on a slide using a Pasteur pipette, which was then analyzed under a microscope after staining with Lugol’s iodine, and the number of *S. haematobium* eggs were counted. 

### 2.5. Data Analysis

Data were analyzed using SAS version 9.4 (SAS Institute, Cary, NC, USA). The results, based on combined UFM + FCM, were used as the primary analytical referent. The theoretical basis for employing combined results as a reference is largely premised on the fact that UFM is an inadequate topological tool for detecting *S. haematobium* infections due to its poor sensitivity for low-intensity infections [10]. Prior studies have successfully managed to use combined results as a reference against which multiple diagnostic devices are compared for their ability to detect soil-transmitted helminths and other parasites [22,23,24,25]. For the purposes of categorizing the children’s infection intensity, the “true” egg count was determined as the larger result from the two testing methods. Children whose largest egg count was 50 eggs per 10 mL of urine or greater were categorized as “heavy intensity”; those whose largest egg count was between one and forty-nine were categorized as “low intensity”; and those with an egg count of zero for both methods were categorized as “uninfected” [26]. Specificity for both UFM and FCM was assumed to be 100% when calculating the sensitivity for both methods.

The prevalence of *S. haematobium* infection was calculated and stratified by age (5–10 and 11–15 years of age) and sex (females and males). Prevalence estimates were calculated by dividing the number of positive cases by the number examined/total study population. The performance of the two testing devices was then calculated against the analytical referent, including sensitivity and negative predictive value, as well as their respective 95% confidence intervals. 

The mean counts of *S. haematobium* eggs per 10 mL of urine were computed for each testing method. To determine statistically significant differences in mean egg counts between the two methods, the Wilcoxon ranked-sign test was performed, and *p*-values were generated. The agreements of the two testing methods at classifying infection status against the referents were calculated using Cohen’s kappa (for infection status). Cohen’s kappa was also calculated for determining the agreement between the two testing methods at classifying infection intensity. For the Cohen’s kappa estimates, values of 0.00–0.20 were interpreted as “no agreement”, values of 0.21–0.39 were interpreted as “minimal agreement”, values of 0.40–0.59 were interpreted as “weak agreement”, values of 0.60–0.79 were interpreted as “moderate agreement”, values of 0.80–0.90 were interpreted as “strong agreement”, and values of 0.91–1.00 were interpreted as “perfect agreement” [27]. Finally, percent agreements between the two testing methods against the referents (for egg counts) were estimated using Spearman’s correlation coefficients due to the non-normal distribution of the data. 

## 3. Results

### 3.1. Prevalence of S. haematobium Infection among Recruited Children

Among the 572 children who provided sufficient urine samples, 17% (*N* = 98) tested positive for *S. haematobium* infection in at least one of the two diagnostic tests, 95% of whom had low-intensity infections (Table 1). Older children (ages 11 through 15) had a greater prevalence of infection (24%, *N* = 31) than the younger ones (15%, *N* = 67). Boys (19%, *N* = 59) were more often infected than girls (15%, *N* = 39). Older children were twice as likely to have heavy-intensity infections than younger children; 3% (*N* = 1) of older children had heavy-intensity infections compared to 6% (*N* = 4) of younger children. However, boys (*N* = 3) and girls (*N* = 2) were equally as likely to have heavy-intensity infections, with 5% of those infected among both sexes being classified as such.

### 3.2. Performance of FlukeCatcher at Detecting S. haematobium Infection

According to the combined UFM + FCM results, 98 urine samples were found to contain *S. haematobium* eggs. Of those, 82 tested positive with FCM (Table 2), while 64 tested positive using UFM. FCM detected more “true positives” and fewer “false negatives” than UFM.

When using the combined results as the analytical referent, FCM outperformed UFM on all metrics at detecting *S. haematobium* eggs in urine. FCM’s sensitivity (84%, 95% CI: 77%–91.0%) was higher than that of UFM (65%, 95% CI: 56%–75%) (Table 3). Its negative predictive value was also higher (97%, 955 CI: 95%–98%) than that of UFM (93%, 95% CI: 91%–96%). In the given context, samples that tested negative with FCM had a greater chance of being a true negative compared to those that tested negative with UFM. Likewise, of those children who were found to excrete *S. haematobium* eggs in their urine, they were more likely to be classified as positive with FCM than with UFM. 

### 3.3. Performance of FlukeCatcher at Estimating Egg Counts

Though FCM had superior sensitivity compared to UFM at detecting infection, it yielded fewer eggs per sample than UFM (Table 4). Among children with positive test results, FCM estimated an average of 6.6 eggs per 10 mL of urine compared to UFM, which estimated an average of 14.7 eggs per 10 mL of urine. Indeed, across all age and infection intensity strata, FCM consistently detected fewer eggs. The differences in mean egg counts between FCM and UFM were more pronounced among older children than younger ones (*p* < 0.0001). This remained true when focusing only on low-intensity infections (*p* < 0.0001). Regardless of which testing method was used, older infected children had considerably greater mean egg counts than younger children. UFM found that older children excreted, on average, 2.6 more eggs per 10 mL of urine than younger ones, whereas FCM estimated a smaller figure with older infected children excreting, on average, 0.7 more eggs per 10 mL of urine than younger ones. Among those with low-intensity infections, FCM detected, on average, 3.9 fewer eggs per 10 mL of urine than UFM, but among those with heavy-intensity infections, FCM detected, on average, 54.6 fewer eggs per 10 mL of urine than UFM. The maximum number of *S. haematobium* eggs in individual urine samples detected using FCM was <49 per 10 mL of urine.

### 3.4. Agreement between FCM and UFM

When classifying infection intensity, UFM and FCM had virtually no agreement (−0.89, *p* = 1.00) (Table 5). FCM failed to identify a single heavy-intensity infection, whilst UFM identified five. FCM and UFM barely had moderate agreement in classifying *S. haematobium* infection status (0.61, 95% CI: 0.51–0.71) (Table 5). However, FCM had a strong agreement with the combined FCM + UFM results at classifying infection status (0.90, 95% CI: 0.84–0.95) (Table 5). In measuring egg counts, FCM had a near perfect correlation with the combined FCM + UFM results (0.91, *p* < 0.0001), whereas UFM had a strong correlation (0.82, *p* < 0.0001). There was a moderate correlation in egg counts between UFM and FCM (0.66, *p* <0.0001). 

## 4. Discussion

This study was undertaken to assess the incremental value of urine filtration using FCM in the detection *S. haematobium* eggs compared to standard UFM. The results of this study confirm that filtering urine through FCM prior to microscopic analysis results in the better detection of *S. haematobium* eggs in human urine samples and catches more low-intensity infections that may otherwise go undetected using UFM. This occurs at the cost of the poorer characterization of the intensity of infection. One advantage that FCM brings is two layers of membranes with two inches of width and mesh sizes of 125 nm and 30 nm, each of which have different pore sizes to catch different sized materials [28], which may potentially increase the likelihood of more effectively catching *S. haematobium* eggs. This, together with supernatant volume reduction, markedly reduces the amount of debris that is seen when the sample is examined under the microscope, resulting in clearer images and easier identification of eggs. Another advantage of FCM is that standard UFM employs a single polycarbonate or nylon membrane that is screwed inside the filtering device, allowing egg loss if the membrane is damaged, improperly placed, or insufficiently sealed.

Paradoxically, FCM detects fewer eggs on average than standard UFM. There was a statistically significant difference in average egg counts among all infected persons, as well as younger children, older children, and those with low-intensity infections. The number of those children with heavy-intensity infections was very small, however. The reason that the incorporation of FCM negatively impacted egg counts is uncertain. The use of FCM resulted in the detection of more low-intensity infections than standard UFM, which may have skewed the mean. It is also possible that the procedure for removing the supernatant from the test tube results in decreased egg counts. Extra care should be taken when processing samples using FCM. Also, there may be challenges in liberating eggs from FCM matrices, particularly when sprayed with water with insufficient pressure.

It is not unprecedented, however, for FCM to have superior sensitivity but also quantify fewer eggs per sample. A previous study found FCM had consistently greater sensitivity but generated smaller egg counts of *Fasciola hepatica* and *Calicophoron daubneyi* among bovine fecal samples [15]. In this study, FCM’s higher sensitivity against Mini-FLOTAC and sedimentation was most pronounced when detecting parasite eggs among cattle with low-intensity infections. However, when it came to quantifying egg counts and classifying infection intensity, FCM was bested by Mini-FLOTAC. FCM does perform better than sedimentation at quantifying egg counts, though [15,29]. It seems, therefore, that FCM is better for diagnosing mammals with helminthic infections, but should be avoided when seeking to classify an individual’s infection intensity if better alternatives exist. 

FCM’s superior agreement with the combined FCM + UFM results at classifying infection status compared to UFM demonstrate that FCM is more in accordance with the “true” results than UFM. Likewise, the same can be said of FCM’s correlation with the average egg counts. As the lowest agreement is found when FCM is compared directly with UFM, it is safe to say that FCM and UFM have significant differences in the results they produce. As Table 5 confirms, UFM and FCM have only moderate agreement with each other. As a result, our study confirms that UFM is overall an insufficient tool at detecting *S. haematobium* infections, particularly low-intensity infections, but adding the two tools together may generate more comprehensive results and cast a wider net in detecting cases of urogenital schistosomiasis. 

Whilst FCM was superior at identifying infected children compared to UFM, both methods still produced considerable amounts of false negatives. Of the 98 children who were found to be excreting any *S. haematobium* egg in their urine, UFM failed to identify 34 of them, whereas FCM failed to identify 16. Indeed, according to the combined results, the overall prevalence of *S. haematobium* infection was 17% among all children. If the UFM results were used as the sole tool for detecting infection, the observed prevalence of *S. haematobium* infection would be 11%, in contrast to 14% with FCM. Therefore, the best value from the use of FCM may be in adding it to existing modalities rather than in replacing them. This echoes a previous study which found that employing additional devices to existing testing methods can increase prevalence rates of Schistosomiasis by improving the overall detection of the parasite [22]. Limited conditional dependence between the compared filtration methods may enhance the ability to yield incremental value by employing both tests [30]. However, despite the superior sensitivity of microscopy when incorporating FCM, it does not seem to be a reliable tool for classifying infection intensity. In our study, it had no agreement with UFM at classifying infection intensities and failed to detect a single heavy-intensity infection. 

FCM and UFM had negative predictive values (NPVs) against the combined FCM + UFM results of 97% and 93%, respectively, supporting FCM’s use in the diagnosis of urogenital schistosomiasis in people. However, NPVs are influenced by prevalence—as prevalence increases, NPVs decrease [31], and our work assessing population surveillance strategies emphasizes low prevalence settings. The calculated prevalence from this study (17%) is lower than that estimated from a previous study on the same population (21%) [18]. If this study were replicated on a population with greater prevalence, the NPV would likely be reduced. The children with discordant results (i.e., one method tested positive while the other tested negative) were more likely to be male (64%) compared to the entire sample population (55%), with a trend to older age. All of those with discordant results were classified as low-intensity infections and were more likely to be identified using FCM. This reflects a similar finding from a previous study which found that discordant results in diagnosing *S. haematobium* were found primarily in persons with egg counts between one and five eggs per 10 mL of urine [32].

The results of this study have implications for schistosomiasis control and elimination efforts. This study demonstrates that processing urine through FCM improves the sensitivity of *S. haematobium* egg detection, catching more low-intensity infections in contrast to standard UFM. It is an easy-to-use and well-tested veterinary product for parasite detection [27,28]. Future research should aim to not only confirm these findings, but also to test FCM for detecting schistosomes in human feces, where eggs of other schistosomes (e.g., *S. mansoni*, *S. japonicum*, *S. mekongi*, *S. intercalatum*) are found. Eggs of different schistosome species have differing shapes and sizes which may interact differently with FlukeCatcher’s membranes. Such research would build upon a previous study which found FCM to be a good alternative for detecting Fasciola eggs in human feces compared to the Kato-Katz thick smear [14], and another study which demonstrated its use in detecting schistosomes in cattle feces [17]. This study is the first to evaluate the performance of the product in human urine, the first to evaluate it for human schistosomiasis, and the second [14] to study its use on humans.

As in the introduction of any innovation into existing workflows, there are potential obstacles to deploying FCM in operational settings. A cost–benefit analysis must be undertaken to determine whether the increased cost in time, personnel effort, and water is justified by the increase in sensitivity. It must also be determined whether FCM has similar performance at isolating other forms of schistosomes in human stool. Further research is needed before it can be recommended for use in clinical diagnostics and population screenings. 

Findings from the present study contribute to our knowledge of schistosomiasis diagnostics by demonstrating how the addition of FCM to UFM can improve the detection of *S. haematobium*. Yet, despite the apparent improvement in performance by combining the testing methods, it is still inadequate at reliably finding individuals with light and ultra-light intensity infections. Previous research has analyzed the addition of microhematuria reagent strips to UFM [32]. It is worth suggesting that future studies should seek to analyze the incorporation of even more diagnostic tools to detect *S. haematobium* in urine, including lateral flow antigen tests [33].

Our study has limitations. First, processing the samples using FCM involved many steps, which, in turn, leads to opportunities for error. Some of these steps are susceptible to inter-site and inter-laboratorian variation, such as the water pressure available for releasing eggs from the matrices. Although laboratory staff took great care to ensure the steps for processing samples with FCM were followed, even the small shaking of the test tubes may have resulted in some eggs being removed from the sample prior to analysis, which might explain why FCM produced significantly smaller egg counts. The study’s dispersed field operations could have also introduced bias (e.g., variability in time intervals between observations at communities driven by operational exigencies). Moreover, some of the participants’ ages may have been misclassified given that various children were unable to recall their age, which then had to be obtained by other villagers. 

Nonetheless, the results of this study show that FCM is a promising tool to better detect low-intensity *S. haematobium* infections. However, its deployment in fieldwork in resource-poor environments requires a clear use case and the assessment of appropriate resources. Opportunities remain to assess the use of the tool for the detection of other schistosome eggs in human stool. 

## Figures and Tables

**Table 1 diagnostics-14-01037-t001:** Prevalence of *S. haematobium* infection among recruited children in Afar, Ethiopia, July 2023.

		Infection Status *			
		None	Light	Heavy	Total
Age (years)	5–10	375 (85%, 95% CI: 81.7%, 88.3%)	63 (14%, 95% CI: 10.8%, 17.2%)	4 (1%, 95% CI: 0.07%, 1.9%)	442
	11–15	99 (76%, 95% CI: 68.7%, 83.3%)	30 (23%, 95% CI: 15.8%, 30.2%)	1 (1%, 95% CI: 0.0%, 2.7%)	130
Sex	Female	216 (85%, 95% CI: 80.6%, 89.4%)	37 (14%, 95% CI: 9.7%, 18.3%)	2 (1%, 95% CI: 0.0%, 2.0%)	255
	Male	258 (81%, 95% CI: 76.7%, 85.3%)	56 (18%, 95% CI: 13.8%, 22.2%)	3 (1%, 95% CI: 0.0%, 2.0%)	317
	Total	474 (83%)	93 (16%)	5 (1%)	572

* Infection status was determined by the greatest egg count of the two testing methods used.

**Table 2 diagnostics-14-01037-t002:** FCM and UFM results compared with combined FCM + UFM results as the reference.

		FCM + UFM Combined Result		Total
		Positive	Negative	
FCM result	Positive	82	0	82
	Negative	16	474	490
UFM result	Positive	64	0	64
	Negative	34	474	508
Total		98	474	572

**Table 3 diagnostics-14-01037-t003:** Performance of FCM and UFM using the combined FCM + UFM results as a reference.

		Combined FCM and UFM Results as Reference (95% CI)
FCM	Sensitivity	84% (77%–91%)
	Negative predictive value	97% (95%–98%)
UFM	Sensitivity	65% (56%–75%)
	Negative predictive value	93% (91%–96%)

**Table 4 diagnostics-14-01037-t004:** Mean egg counts (number of eggs per 10 mL of urine) among infected children by testing method ^1^.

		Testing Method		*p*-Value ^3^
		UFM	FCM	
Age (years)	5–10	13.8 (SD = 22.6)	6.4 (SD = 7.3)	<0.0001
	11–15	16.4 (SD = 18.7)	7.1 (SD = 8.5)	<0.0001
Infection intensity ^2^	Light intensity	9.8 (SD = 12.8)	5.9 (SD = 7.1)	<0.0001
	Heavy intensity	72.4 (SD = 13.4)	17.8 (SD = 7.7)	0.0625
	All infected	14.7 (SD = 21.2)	6.6 (SD = 7.7)	<0.0001

^1^ Values of zero were excluded from calculations. ^2^ Participants’ infection intensity was determined by the larger of the counts obtained using the two testing methods. This is why the average FCM egg count for heavy-intensity infections is less than 50. ^3^
*p*-values were calculated using Wilcoxon signed-rank test.

**Table 5 diagnostics-14-01037-t005:** Agreement between FCM and UFM against each other as well as against the combined FCM + UFM results in diagnosing infection and classifying infection intensity, and correlation with egg counts.

Tests Compared	Cohen’s Kappa for Classifying Infection Status (95% CI)	Spearman’s Correlation for Estimating Egg Count (*p*-Value)	Cohen’s Kappa for Classifying Infection Intensity (*p*-Value)
UFM vs. combined results	0.76 (0.68–0.83)	0.82 (*p* < 0.0001) *	-
FCM vs. combined results	0.90 (0.84–0.95)	0.91 (*p* < 0.0001) *	-
FCM vs. UFM	0.61 (0.51–0.71)	0.66 (*p* < 0.0001)	−0.89 (*p* = 1.00)

* Average egg count between the two testing methods was treated as the “combined results”.

## Data Availability

The data presented in this study are available on request from the corresponding author. The data are not publicly available due to privacy/ethical issues.

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
