# Peer review of "Evaluating the Performance of FlukeCatcher at Detecting Urogenital Schistosomiasis"

_diagnostics, 2024, doi:10.3390/diagnostics14101037_

Round 1

Reviewer 1 Report

Comments and Suggestions for Authors

The article is well written and shows an important application of an existing tool to improve detection of urinary schistosomiasis. Please see my comments related to the improvement of the manuscript.

Lines 65-69: The use of urine dipstick can improve detection of infection and not of egg detection (as stated in the submission). Please revise accordingly.

Line 85-87: Please indicate sampling design of the larger study and basis for sample size of the current study. If results of larger study have been published, please include in the citation.

Line 92: Indicate what is the lower limit for assent in this study.

Line 144: Suggest to change analytical referents to composite reference standards. Also, please justification why include the UFM alone as reference standard and not just the composite UFM+FCM. Indicate also if specificity is assumed to be 100% for both UFM and FCM (given the expertise of the microscopists to ID the eggs). 

Line 177-182 and Table1: Suggesting to present prevalence with 95% CI.

Line 189-219 and Tables 2-3: Suggesting to just include the results for the composite reference standard as the information of UFM alone as reference standard can be presented as UFM vs the composite reference standard. With this, suggesting also to just use Table 3 and add in the row UFM result.

Table 4: Suggesting to just use composite reference standard.

Author Response

Dear Reviewer 1,

We are grateful for your careful review and constructive comments, which have helped to improve the manuscript. We have made changes to the manuscript based on your suggestions and describe these changes in the below paragraphs. We hope that you will find our responses acceptable, and we look forward to your decision.

  1. Lines 65-69: The use of urine dipstick can improve detection of infection and not of egg detection (as stated in the submission). Please revise accordingly.

  • We have revised the sentences in lines 65-69 as suggested. The revised text reads as “Current diagnostic modalities of haematobium can be improved by adding simple technologies to better detect infections. For instance, the use of urine dipsticks in conjunction with standard microscopy can improve the sensitivity of infection detection.”

  1. Line 85-87: Please indicate sampling design of the larger study and basis for sample size of the current study. If results of larger study have been published, please include in the citation.
  • We have added text explaining briefly the study design for the larger study as well as the sample size estimation technique (See line 92-107). It reads

  • Study design

This study was conducted as a cross-sectional validation study. The research protocol was part of a larger study that examined the performance of pooled testing in detecting and estimating the intensity of S. haematobium infection. The larger pooling study involved examination of individual and pooled urine samples collected from school-age children (5 to 15 years) living in the Afar, Gambela, and Benishangul Gumuz regional states of Ethiopia, where there is S. haematobium transmission [3,18]. Urine samples were collected before and one month after praziquantel treatment and examined for S. haematobium infection using urine filtration microscopy.

  • Sample size determination

Because there is insufficient data regarding the sensitivity of FC in detecting S. haematobium eggs in urine samples, we relied on the sensitivity reported in a previous study (>90% when egg per gram was >20) for detecting F. hepatica in bovine fecal samples to estimate the sample size [15]. A previous similar study reported that the prevalence of S. haematobium in the current study area (Middle Awash area) is 28.6% [18]. Assuming a margin of error 0.05, Zα/2=1.96 and 15% non-response rate, the samples size estimated to determine the sensitivity of FC in detecting S. haematobium infection in human urine sample was 558 [19].

  1. Line 92: Indicate what is the lower limit for assent in this study.

  • We have provided a text describing the age limit for assent in this study. It reads “All children participated in this study with ages from 5 to 15 years were eligible for assent” (see pages 114 & 115).

  1. Line 144: Suggest changing analytical referents to composite reference standards. Also, please justification why includes the UFM alone as reference standard and not just the composite UFM+FCM. Indicate also if specificity is assumed to be 100% for both UFM and FCM (given the expertise of the microscopists to ID the eggs). 

  • Text in the data analysis and results which are based on UFM alone as reference have been removed.

  • We have stated that the specificity was assumed 100% for UFM and FCM. It reads “Specificity for both UFM and FCM was assumed 100% when calculating the sensitivity for both methods” (lines #177 & 178).

  1. Line 177-182 and Table1: Suggesting to present prevalence with 95% CI.

  • We have provided 95% CI for the values in Table 1

  1. Line 189-219 and Tables 2-3: Suggesting to just include the results for the composite reference standard as the information of UFM alone as reference standard can be presented as UFM vs the composite reference standard. With this, suggesting also to just use Table 3 and add in the row UFM result.

  • We have presented the results based on only the combined UFM and FCM as a reference. We have added the results of UFM vs the combined UFM and FCM in Table 3 and deleted Table 2.

  1. Table 4: Suggesting to just use composite reference standard.

  • The UFM reference column has been removed from Table 4 (Table 3 in the revised manuscript). Only the composite reference standard is presented.

Reviewer 2 Report

Comments and Suggestions for Authors

The review is interesting, useful and has practical importance for the study area. But some adjustments need to be made, such as:

Line 26. It is not clear what the abbreviation means “The sensitivity and NPV of detecting S. Haem….” 

Lines 162-164. The sentence is not clear „The average urine egg counts (UEC, expressed in eggs per 10 mL of urine – EPU) for each testing method amongst positive samples were also calculated.“

Line 185. Table 1: prevalence of S. haematobium infection among recruited children in Afar,……” The sentence should start with a capital letter - Prevalence…..

Line 190 and 191. „48 of 64 urine samples found to contain S. haematobium eggs according to UFM ….“ The sentence should start with words.

Line 200. “Table 2: comparing the performance of FCM i….” The sentence should start with a capital letter.

Line 234. The sentence should start with a capital letter.

Line 46  ..................... parasitic infection worldwide after malaria [6, p.]. What does it mean - p.?

More data on the FlukeCatcher method such as figure would be helpful.

Data if available on eosinophilia in children would be interesting and would be useful for diagnosis.

Author Response

Dear Reviewer 2,

We are grateful for your careful review and constructive comments, which have helped to improve the manuscript. We have made changes to the manuscript based on your suggestions and describe these changes in the below paragraphs. We hope that you will find our responses acceptable, and we look forward to your decision.

  1. Line It is not clear what the abbreviation means “The sensitivity and NPV of detecting S. Haem….” 
  • On page 1 (abstract), the term for “NPV” has been written in full (See line 26).
  1. Lines162-164. The sentence is not clear „The average urine egg counts (UEC, expressed in eggs per 10 mL of urine – EPU) for each testing method amongst positive samples were also calculated.“
  • Thank you. We have revised the sentence to improve clarity. It reads “The mean counts of haematobium eggs per 10 mL of urine was computed for each testing method” (see line 186 & 187).
  1. Line Table 1: prevalence of S. haematobium infection among recruited children in Afar,……” The sentence should start with a capital letter - Prevalence…..
  • The heading for table 1 starts with a capital letter.
  1. Line 190 and 191.„48 of 64 urine samples found to contain S. haematobium eggs according to UFM ….“ The sentence should start with words.
  • On page 5, this sentence has been removed per feedback from reviewer 1.
  1. Line 200. “Table 2: comparing the performance of FCM i….” The sentence should start with a capital letter.
  • On page 5, this sentence has been removed per feedback from reviewer 1.
  1. Line 234.The sentence should start with a capital letter.
  • On page 6 (table 4), the heading for table 3 (table 4 in the older version) starts with a capital letter.
  1. Line 46  ..................... parasitic infection worldwide after malaria [6, ]. What does it mean - p.?
  • On page 1 (introduction), the “-p” has been removed. It was originally added automatically by the citation software for unknown reasons.
  1. More data on theFlukeCatcher method such as figure would be helpful.
  • We have added data/summarized findings from literature regarding the performance of FlukeCatcher in detecting Flukes in fecal samples.

The added text reads as follows “Compared to Mini-FLOTAC and sedimentation technique, FlukeCatcher consistently showed the highest sensitivity levels across all infection intensities of Fascioliasis in bo-vine fecal samples [15]. Likewise, FlukeCatcher detected all fecal samples from sheep and cattle with Fasciola hepatica eggs [16]. The FlukeCatcher also quantified more eggs than the Becker method, and showed a significantly lower egg detection threshold [16]. This level of success can be also seen in the detection of schistosomes in cattle [17]. As a result, FlukeCatcher holds “promising potential” [16] in enhancing detection in liver flukes, es-pecially schistosomes, in human-derived samples.” (lines 77-84).

  1. Data if available on eosinophiliain children would be interesting and would be useful for 

  • We share the idea that data on eosinophiliain children would be useful. However, we didn’t collect data on eosinophilia to report in the current study.   

Round 2

Reviewer 1 Report

Comments and Suggestions for Authors

The revised manuscript is now acceptable for publication.